# Infectious dose of *Senecavirus A* in market weight and neonatal pigs

**Alexandra Buckley***, **Kelly Lager**

Virus and Prion Research Unit, National Animal Disease Center, Agriculture Research Service, United State Department of Agriculture, Ames, Iowa, United States of America

* alexandra.buckley@usda.gov

**Data Availability Statement:** All relevant data are within the paper and its Supporting Information files.

**Funding:** The author(s) received no specific funding for this work.

## Abstract

Foot-and-mouth disease virus (FMDV) is a picornavirus that produces a highly transmissible vesicular disease that can devastate meat and dairy production to such an extent that FMDV-free countries commit significant economic resources to maintain their FMDV-free status. *Senecavirus A* (SVA), also a picornavirus, causes vesicular disease in swine that is indistinguishable from FMDV. Since 2015, SVA outbreaks have been reported around the world requiring FMDV-free countries to investigate these cases to rule out FMDV. Understanding the pathogenesis of the SVA and its ability to transmit to naïve populations is critical to formulating control and prevention measures, which could reduce FMDV investigations. The primary objective of this study was to determine the infectious dose of SVA in market weight and neonatal pigs. A 2011 SVA isolate was serially hundred-fold diluted to create four challenge inoculums ranging from $10^{6.5}$ to $10^{0.5}$ TCID$_{50}$/ml. Four market weight pigs individually housed were intranasally inoculated with 5 mL of each dose (n = 16). Serial ten-fold dilutions were used to create 6 challenge inoculums ranging from $10^{5.5}$ to $10^{0.5}$ TCID$_{50}$/ml for neonatal pigs. Again, four animals in individual housing were challenged orally with 2 mL of each dose (n = 24). Detection of SVA by PCR in collected samples and/or neutralizing antibody response was utilized to classify an animal as infected. The minimum infectious dose for this study in market weight animals was 1,260 TCID$_{50}$/ml ($10^{3.1}$ TCID$_{50}$/ml) and for neonates it was 316 TCID$_{50}$/ml ($10^{2.5}$ TCID$_{50}$/ml). Knowledge of the infectious dose of SVA can guide biosecurity and disinfection measures to control the spread of SVA.

## Introduction

*Senecavirus A* (SVA) is a small, non-enveloped, single-stranded, positive-sense RNA virus in the genus *Senecavirus* and family *Picornaviridae* [1]. It was originally discovered as a cell culture contaminant in Maryland in 2002 and named Seneca Valley virus (SVV-001); although, similar picorna-like viruses had been identified in United States (US) swine samples dating back to the late 1980s [2, 3]. Early experimental inoculation of pigs resulted in viral replication and an antibody response but did not produce any specific clinical disease [2, 4]. In late 2014, vesicular disease outbreaks including increased neonatal mortality on sow farms occurred in Brazil and similar cases were reported in the US in 2015 with diagnostic testing confirming the

**Competing interests:** The authors have declared that no competing interests exist.

presence of SVA [5–8]. Evidence from the field and experimental reproduction of vesicular lesions in pigs from weaning to breeding age demonstrated that SVA was a causative agent for vesicular disease in swine [9–12].

Understanding the epidemiology of SVA and ways to prevent spread is important considering it causes clinical disease that is grossly indistinguishable from foot-and-mouth disease virus (FMDV). FMDV is an economically devastating disease not only for FMDV-free countries with an introduction event, but also for those countries with endemic virus battling production losses and vaccination costs [13]. Since FMDV is on the World Organization for Animal Health (OIE) list of notifiable diseases, having an endemic vesicular disease in a country free of FMDV also costs time and money since every time a vesicular lesion is observed it must be investigated to rule out FMDV [14].

In addition to vesicular disease, both Brazil and the US reported increases in neonatal mortality on affected farms with clinical signs including lethargy, wasting, and occasionally diarrhea, but Brazil also reported vesicular lesions and neurologic signs in neonates not commonly observed in the US [15–17]. Further testing in many cases ruled out other causative agents for the clinical signs observed, thus leading to the conclusion that SVA was also the cause of mortality in piglets on affected breeding farms. Currently, there have been no published reports of experimental reproduction of neonatal morbidity or mortality with SVA.

Infectious virus has been isolated from environmental samples on swine farms including dust from fans, hallways in barns, and even the ground outside barns [18]. In addition, SVA has been isolated from environmental samples taken from an assembly yard for animals prior to shipment for slaughter [19]. Not only has SVA been found on environmental surfaces, but also in mouse feces and flies providing potential vectors [18]. Demonstration of virus in the environment by both PCR and virus isolation has led to the question of whether this material could be infectious to pigs and play a role in transmission.

Current environmental testing by PCR for SVA can be difficult to interpret due to the lack of information about the infectious dose of SVA. The primary objective of this study was to determine the minimum infectious dose (MID) and infectious dose 50% (ID50) of SVA in both market weight and neonatal pigs. An additional goal was evaluating the correlation between PCR Ct values, tissue culture infectious dose (TCID), and infectivity in swine. Results from this study could provide context for interpreting SVA PCR results especially those collected from environmental samples.

## Materials and methods

### Cells and virus

Swine testicular (ST) cells received from the National Veterinary Services Laboratory in Ames, Iowa were grown in minimum essential medium (MEM, MilliporeSigma, St. Louis, MO) supplemented with 10% fetal bovine serum (FBS, AtlantaBio, Flowery Way, GA), 1% L-glutamine (Life Technologies, Carlsbad, CA), and gentamicin at 37°C and 5% $CO_2$. ST cells were used for virus growth, titration, and virus neutralization assays. The virus used for animal inoculation was supplied by the National Veterinary Services Laboratory (strain 11-55910-3). It was isolated from a swine brain sample collected from a group of market hogs traveling to slaughter from Canada to the US that presented with lameness and vesicular disease (GenBank KC667560). The isolate was propagated on ST cells with the second and third passage being pooled and diluted to a titer of $1\times10^{6.5}$ TCID50/mL, which was termed the stock virus (SVA/CAN/2011).

## Inoculum and virus titration

Inoculum for the market weight pigs was made by serial hundred-fold dilutions of the stock virus (SVA/CAN/2011) with MEM to create four challenge inoculums ranging from $10^{6.5}$ to $10^{0.5}$ $TCID_{50}$/mL. Inoculum for the neonates was made by serial ten-fold dilutions of SVA/CAN/2011 with MEM to create 6 challenge inoculums ranging from $10^{5.5}$ to $10^{0.5}$ $TCID_{50}$/mL.

Virus titrations were performed on confluent ST cells in a 96-well plate with each well containing 100uL of growth medium. Virus stocks were serially ten-fold diluted in MEM and 100uL was added to seven wells in the column of a plate with the remaining well serving as a control for each dilution. Plates were incubated and examined microscopically for cytopathic effect for 4 days after inoculation. Viral titers were calculated using the Reed and Muench method [20].

## Animal study design

All animal research was performed in accordance with an Animal Care and Use Protocol (ACUP ARS-2867) approved by the National Animal Disease Center's (NADC) Institutional Animal Care and Use Committee. At the end of study, all animals were humanely euthanized with the intravenous administration of a barbiturate (Fatal Plus, Vortech Pharmaceuticals, Dearborn, MI) following the label dose (1 mL/4.45 kg).

Study 1: Sixteen pigs born and raised at the NADC ranging from eight to nine months-of-age were randomly divided into 4 treatment groups (n = 4/group): $10^{6.5}$, $10^{4.5}$, $10^{2.5}$, and $10^{0.5}$ $TCID_{50}$/mL inoculum. During the two-weeks animals were on study, they were individually housed in four ABSL-2 rooms. The same four rooms were used for each treatment group with cleaning, disinfection, and drying for 48–96 hours in between groups. Environmental swabs were collected from the top of gating above pig level weekly to monitor SVA in the environment. The $10^{0.5}$ treatment group was challenged first followed by successively higher titer inoculum groups ending with the $10^{6.5}$ treatment group. Pigs were challenged intranasally with 5 mL of inoculum split between both nostrils on 0 days post inoculation (dpi). Pigs were bled on 0, 4, 7, and 14 dpi and rectal swabs were collected daily from 0–14 dpi. Pigs were visually assessed for the formation of vesicular lesions on the coronary bands or snouts daily.

Study 2: Thirty piglets were farrowed at NADC from sows purchased from a commercial source. Sows were clinically free of vesicular lesions and were negative for neutralizing antibodies against SVA. At 24–72 hours after birth, piglets were weaned from the sow and blocking by litter were randomly allocated to 7 treatment groups. Six treatment groups (n = 4/group) were inoculated with $10^{5.5}$, $10^{4.5}$, $10^{3.5}$, $10^{2.5}$, $10^{1.5}$, or $10^{0.5}$ $TCID_{50}$/mL inoculum respectively. The seventh group (n = 6) of pigs served as room contamination sentinels. Six ABSL-2 rooms, one for each challenge dose, were utilized for the study each containing five isolators and a raised deck (Fig 1A). At weaning piglets were ear notched, bled, oral swabbed and rectal swabbed prior to placement in an individual isolator (-1 dpi) (Fig 1B). Pigs were given 24 hours to acclimate to feeding bowls and milk replacer. During feeding, piglets were fed one at a time by opening the feeding port, pouring in milk replacer closing the port before moving to the next isolator. The sentinel pig was always fed last. Piglets were challenged orally with 2 mL of inoculum on 0 dpi and thereafter isolators were only opened to provide milk replacer through the feeding port. On 6 dpi, piglets were bled, oral swabbed and rectal swabbed as they were individually removed from the isolators (6 dpi) and placed as a group (n = 5) onto the raised deck in the room. Serum and swabs were also collected on 10 and 14 dpi.

All animals were observed daily for clinical signs including lameness, lethargy, inappetence, and diarrhea. Serum was harvested from serum separator tubes (BD Vacutainer®, Franklin Lakes, NJ) used to collect blood. Oral, rectal, and environmental swabs were collected using a

**A**

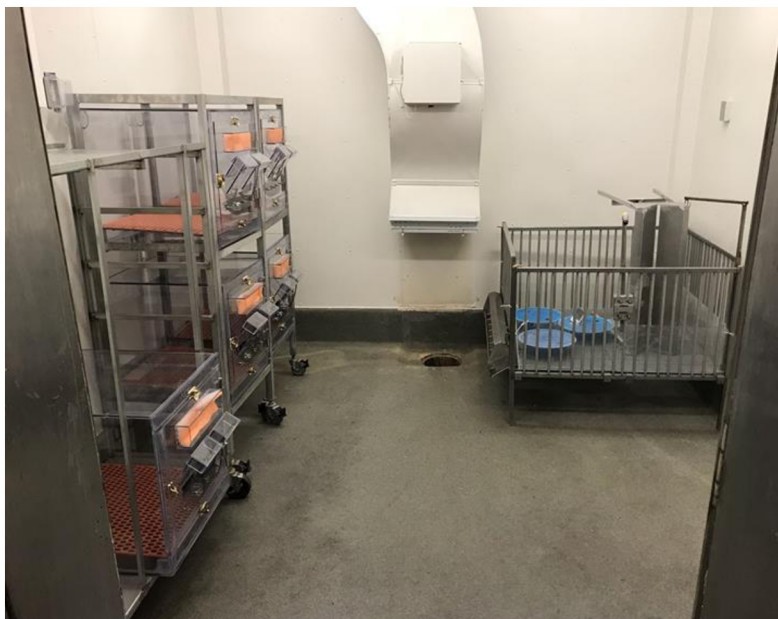

**B**

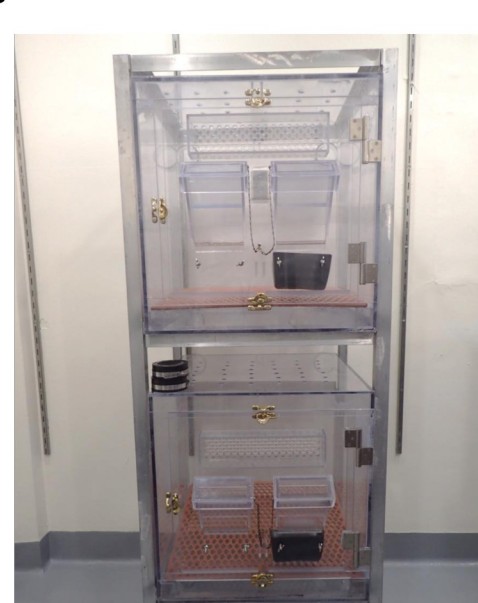

**Fig 1. Housing for the neonatal pig infectious dose study.** A) Room set up for each challenge dose. Five individual housing units with a raised deck for subsequent group housing. B) Each individual unit has a small port used for feeding and holes in the unit for air circulation were covered with filter paper.

sterile polyester tipped applicator (Puritan Medical Products, Guilford, ME), immersed in 3 mL of MEM. Samples were stored in a -80˚C freezer for future testing.

### SVA nucleic acid extraction and quantification

Inoculum, serum, and swabs were tested by real-time reverse transcriptase-polymerase chain reaction (RT-qPCR) as previously described [21]. Briefly, RNA was extracted from samples using the MagMAX™ Pathogen RNA/DNA kit (Applied Biosystems, Foster City, CA) following kit instructions for sample type. Next, 5 μL of extracted product was added to 20 μL of the Path-ID™ Multiplex One-Step RT-PCR reaction master mix for fecal swabs or 20 μL AgPath-ID™ One step RT-PCR master mix (Applied Biosystems) for inoculum, sera, and oral swabs. Inoculum samples were tested in triplicate, while serum and swab samples were tested in duplicate. The forward primer sequence was 5'-TGCCTTGGATACTGCCTGATAG-3', the reverse primer sequence was 5'-GGTGCCAGAGGCTGTATCG-3' and the probe sequence was 5'-CGACGGCCTAGTCGGTCGGTT-3'. $C_t$ values greater than 35 for animal samples were considered negative.

A plasmid containing the target region for the primers and probe was used as a standard for quantification. A 490 nucleotide region (313–803) from the 5' untranslated region (5'UTR) and protein L of SVV-001 was cloned into the plasmid vector pCI-neo between the restriction enzymes XhoI and MluI (Promega, Madison, WI). The plasmid was transformed into One Shot® Electrocomp™ cells (Invitrogen, ThermoFisher Scientific) for plasmid propagation on LB agar plates with 50μg/mL Kanamycin. A QIAGEN® Plasmid Maxi Kit was used according to manufacturer's instructions for plasmid purification. RNA was quantified with a NanoDrop spectrophotometer (ThermoFisher Scientific) and the average nucleic acid quantity from three readings was used to calculate genomic copies (GC).

## Virus neutralization (VN) assay

The virus neutralization assay has previously been described [21]. In brief, serum samples were heat-inactivated and serially diluted 1:4 (up to 1:4096) in MEM with four replicates. An equal volume of SVA/CAN/2011 diluted to ~200 $TCID_{50}$ was added to the serum and incubated for 1 hour. The virus-serum mixture was transferred to confluent 96-well plates of ST cells. Plates were microscopically evaluated for cytopathic effect (CPE) daily for 4 days. Titers were recorded as the reciprocal of the highest dilution of serum at which the CPE of the SVA isolate was not visible in 50% of the inoculated wells. Titers ≤16 were considered negative.

## Results

In Study 1, the SVA isolate SVA/CAN/2011 was serially hundred-fold diluted to create four challenge inoculums for finishing pigs with theoretical titers ranging between $10^{6.5}$ to $10^{0.5}$ $TCID_{50}$/mL for the $10^0$ to $10^{-6}$ dilutions respectively (Table 1). Table 1 shows the back-titration results from the inoculum as well as the Ct value and GC/mL from RT-qPCR. Dilutions from $10^0$ to $10^{-4}$ had higher titers than the theoretical titer. The $10^{-6}$ dilution was undetected by $TCID_{50}$ assay. Inoculum samples were tested in triplicate and the Ct value and GC/mL were calculated as an average of the three wells. Inoculum Ct values ranged from 11.0–32.6 which corresponded to $9.99 \times 10^9$ to 4080 GC/mL.

Environmental swabs were collected from gating weekly in the individual animal rooms in Study 1 to provide information about the cleaning and disinfection procedure used in between challenge groups as well as about environmental load of SVA. All environmental samples were PCR negative for SVA nucleic acid until the final group of pigs ($10^{6.5}$). At 7 dpi there was 1 pig room that had a PCR positive sample, but the room was negative the following week (14 dpi).

All four pigs in the $10^{6.5}$ treatment group replicated and shed SVA (Table 2). Rectal swabs were positive for SVA starting on 1 dpi with peak genomic copies observed on 6 dpi (Fig 2A). All animals had multiple positive swabs throughout the study with an average of 7 positive swabs per animal. In contrast, the $10^{4.5}$ treatment group did not have positive rectal swabs until 4 dpi with a similar peak at 6 dpi, but quickly declined. Animals in this group only had 2 positive swabs on average. Both the $10^{6.5}$ and $10^{4.5}$ groups had a smaller second peak of viral detection at 10 and 9 dpi respectively. One animal in the $10^{2.5}$ group had a positive rectal swab on 7 and 8 dpi. No animals in the $10^{0.5}$ challenge group had a positive rectal swab.

**Table 1. Titers, PCR Ct values, and genomic copies/mL of virus dilutions used to inoculate finishing pigs.**

| Inoculum | Dilution of stock virus | $TCID_{50}$/mL (theoretical) | $TCID_{50}$/mL (back titrated) | Ct | Genomic copies/mL |
|---|---|---|---|---|---|
| $10^{6.5}$ | $10^0$ | 3,160,000 | 5,010,000 | 11.0 | 9.99E+9 |
| $10^{4.5}$ | $10^{-2}$ | 31,600 | 63,100 | 18.7 | 5.35E+7 |
| $10^{2.5}$ | $10^{-4}$ | 316 | 1,260 | 25.6 | 4.75E+5 |
| $10^{0.5}$ | $10^{-6}$ | 3.16 | 0 | 32.6 | 4.08E+3 |

**Table 2. Summary of finishing pig PCR positive status for each sample type throughout the study and VN titer for 14 dpi serum.**

| Inoculum | Serum | Oral swabs | Rectal swabs | VN titer |
|---|---|---|---|---|
| $10^{6.5}$ | 4/4 | 4/4 | 4/4 | 1024, 1024, 1024, 4096 |
| $10^{4.5}$ | 2/4 | 2/4 | 4/4 | 1024, 1024, 1024, 4096 |
| $10^{2.5}$ | 1/4 | 0/4 | 1/4 | ≤16, ≤16, 64, 1024 |
| $10^{0.5}$ | 0/4 | 0/4 | 0/4 | ≤16, ≤16, ≤16, ≤16 |

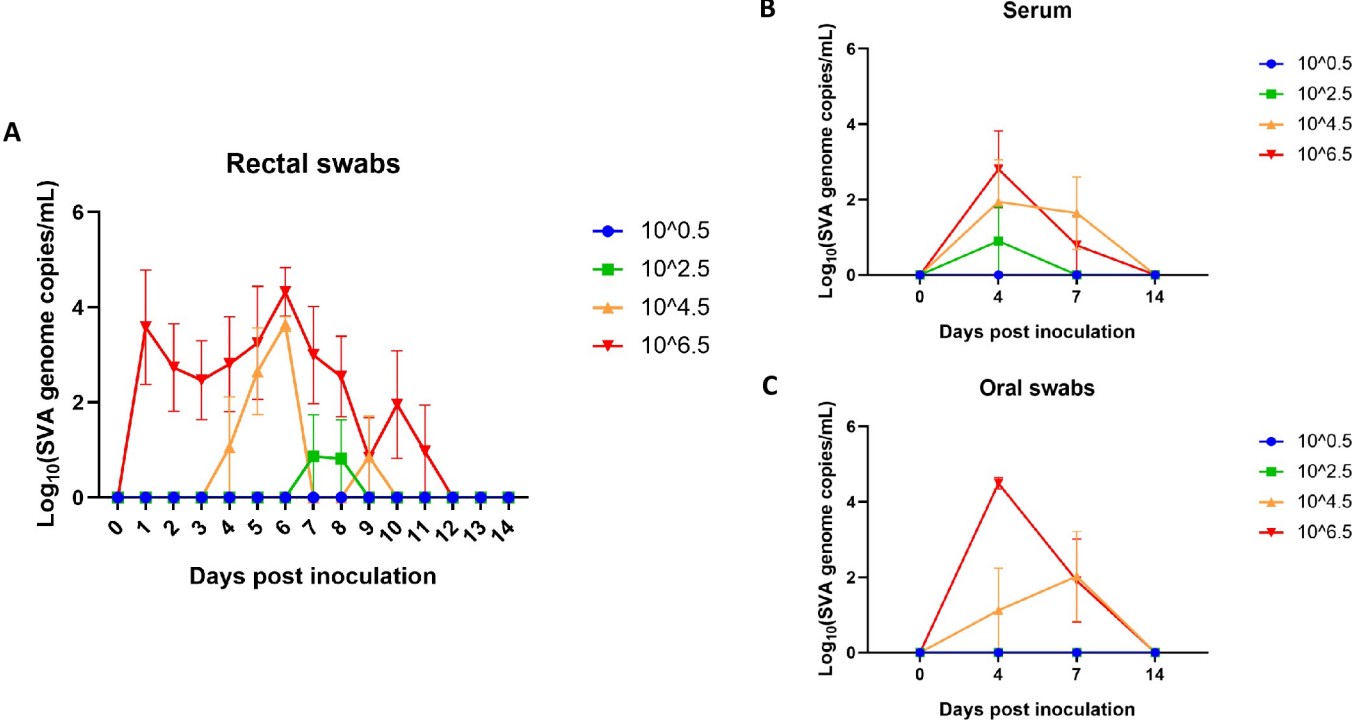

**Fig 2.** SVA infection dynamics measured by RT-qPCR in A) rectal swabs, B) serum, and C) oral swabs collected during the study. The legend provides a key to the color of each inoculum group: $10^{6.5}$ (red), $10^{4.5}$ (orange), $10^{2.5}$ (green), and $10^{0.5}$ (blue). Points on the graph represent the mean of the four animals and error bars and standard error of the mean.

One animal from the $10^{6.5}$ group did not have a detectable viremia, but the other three animals had peak SVA RNA levels at the 4 dpi serum sample (Fig 2B). Two animals in the $10^{4.5}$ group and only one animal in the $10^{2.5}$ group had a detectable viremia (Table 2). Oral swabs taken on the same days as the blood samples had similar PCR results to the serum except that all four animals in the $10^{6.5}$ group had positive oral swabs, while no animals in the $10^{2.5}$ group tested positive (Fig 2C). The same two pigs in the $10^{4.5}$ group had positive serum and oral swabs. Finally, no animals in the $10^{0.5}$ treatment group had positive PCR results for serum or oral swabs.

Virus neutralization titers were determined for 0 dpi and necropsy serum (14–15 dpi). All animals had titers ≤16 on 0 dpi. All animals from groups $10^{6.5}$ and $10^{4.5}$ developed neutralizing titers ranging from 1024–4096 (Table 2). Two pigs in the $10^{2.5}$ groups had titers of 64 and 1024 respectively, while the remaining two pigs had titers ≤16. No pigs in the highest dilution ($10^{0.5}$) seroconverted.

In Study 2, SVA/CAN/2011 was serially ten-fold diluted to create six challenge inoculums for neonatal pigs with theoretical titers ranging between $10^{5.5}$ to $10^{0.5}$ TCID$_{50}$/mL for the $10^{-1}$ to $10^{-6}$ dilutions respectively (Table 3). The $10^{-1}$ to $10^{-3}$ dilutions had higher titers than the theoretical titer, but $10^{-2}$, $10^{-4}$, and $10^{-5}$ were the same as predicted. Again the $10^{-6}$ dilution was undetected by TCID$_{50}$ assay. Inoculum $C_t$ values ranged from 17.4–35.5 corresponding to $1.28 \times 10^8$ to 570 GC/mL.

Due to the piglets being moved from individual animal housing to group housing on a deck, the 6 dpi samples collected as the animal was taken out of individual housing were the only samples used for PCR determination of infection status. In addition, the VN assay was

**Table 3. Titers, PCR Ct values, and genomic copies/mL of virus dilutions used to inoculate neonatal pigs.**

| Inoculum | Dilution of stock virus | TCID$_{50}$/mL (theoretical) | TCID$_{50}$/mL (back titrated) | Ct | Genomic copies/mL |
|---|---|---|---|---|---|
| $10^{5.5}$ | $10^{-1}$ | 316,000 | 501,000 | 17.4 | 1.28E+8 |
| $10^{4.5}$ | $10^{-2}$ | 31,600 | 31,600 | 20.9 | 1.13E+7 |
| $10^{3.5}$ | $10^{-3}$ | 3,160 | 5,010 | 25.1 | 6.39E+5 |
| $10^{2.5}$ | $10^{-4}$ | 316 | 316 | 28.7 | 5.75E+4 |
| $10^{1.5}$ | $10^{-5}$ | 31.6 | 31.6 | 32.3 | 5.12E+3 |
| $10^{0.5}$ | $10^{-6}$ | 3.16 | 0 | 35.5* | 5.70E+2 |

* One of the three wells was undetermined so the value is an average of two wells.

performed on 10 dpi serum assuming piglets infected from pen mates once combined would not have enough time to develop a robust neutralizing antibody response. All four pigs in the $10^{5.5}$ treatment group were viremic, had detectable SVA in oral and rectal swabs, and seroconverted (Table 4). Only 2 pigs in the $10^{4.5}$ treatment group had SVA positive serum and swab samples and a neutralizing antibody response. Both the $10^{3.5}$ and $10^{2.5}$ treatment groups only had 1 pig that had SVA positive samples; although, the pig in the $10^{3.5}$ group had a neutralizing antibody response while the PCR positive pig from the $10^{2.5}$ group did not. All pigs in the $10^{1.5}$ and $10^{0.5}$ treatment groups tested negative for SVA in 6 dpi samples and did not seroconvert. In addition, all the room control pigs were negative for SVA when removed from individual housing.

Samples were also collected on 10 dpi and 14 dpi from all piglets while group housed to observe infection dynamics within each inoculum group. All piglets including room controls from the $10^{5.5}$, $10^{4.5}$, $10^{3.5}$, and $10^{2.5}$ treatment groups were PCR positive in either serum or rectal swabs collected on 10 dpi. By 14 dpi, viremia was detected in less animals, although most animals in these groups still had positive rectal swabs. All piglets in the $10^{1.5}$ and $10^{0.5}$ treatment groups including room controls remained negative.

Tables 5 and 6 summarize the inoculum data with pig infection data. In market weight pigs inoculated with hundred-fold dilutions of SVA, the ID50 and MID was 1,260 TCID$_{50}$/mL (6,300 TCID$_{50}$/pig). In this group, SVA nucleic acid was only detected in samples from one pig, but a neutralizing antibody response was detected in two pigs. The inoculum had a PCR Ct value of 26.5 and the following dilution that did not infect pigs had a Ct value of 32.6. In neonates inoculated with ten-fold dilutions of SVA, the ID50 was 31,600 TCID$_{50}$/mL (63,200 TCID$_{50}$/pig) and the MID was 316 TCID$_{50}$/mL (632 TCID$_{50}$/pig). At the MID only one pig had PCR positive samples and had not developed a neutralizing antibody response by 10 dpi, the remaining piglets in the group became infected when the piglets were co-mingled. This inoculum had a PCR Ct value of 29.7 while the following dilution that did not infect pigs had a value of 33.2.

**Table 4. Summary of neonate PCR positive status for each sample type on 6 dpi and VN titer for 10 dpi serum.**

| Inoculum | Serum | Oral swabs | Rectal swabs | VN titer |
|---|---|---|---|---|
| $10^{5.5}$ | 4/4 | 4/4 | 4/4 | 64, 256, 256, 1024 |
| $10^{4.5}$ | 2/4 | 2/4 | 2/4 | ≤16, ≤16, 256, 1024 |
| $10^{3.5}$ | 1/4 | 1/4 | 1/4 | ≤16, ≤16, ≤16, 1024 |
| $10^{2.5}$ | 1/4 | 1/4 | 1/4 | ≤16, ≤16, ≤16, ≤16 |
| $10^{1.5}$ | 0/4 | 0/4 | 0/4 | ≤16, ≤16, ≤16, ≤16 |
| $10^{0.5}$ | 0/4 | 0/4 | 0/4 | ≤16, ≤16, ≤16, ≤16 |

**Table 5. Summary of SVA infection outcomes in finishing pigs by PCR and VN assays.**

| TCID$_{50}$/mL | TCID$_{50}$/pig* | Ct | PCR positive | VN positive |
|---|---|---|---|---|
| 5,010,000 | 25,550,000 | 11.0 | 4/4 (100%) | 4/4 (100%) |
| 63,100 | 315,500 | 18.7 | 4/4 (100%) | 4/4 (100%) |
| 1,260 | 6,300 | 25.6 | 1/4 (25%) | 2/4 (50%) |
| 0 | 0 | 32.6 | 0/4 (0%) | 0/4 (0%) |

* Pigs were inoculated with 5 mL

## Discussion

The primary goal of this study was to determine the infectious dose of SVA in both market weight and neonatal pigs. Due to the size of market weight pigs and the nature of the study, only one pig could be housed in a room at a time; therefore, due to animal space constraints only 4 dilutions were tested. The size of neonates and the ability to individually house all piglets for one dilution in the same room allowed for a greater range of dilutions to be tested, 6 ten-fold dilutions. Therefore, even though the MID of market weight pigs (1,260 TCID$_{50}$/mL) was a higher titer than the MID of neonates (316 TCID$_{50}$/mL), it does not necessarily mean that neonatal pigs have a lower threshold. This study has helped narrow the window of the MID in finishing pigs and further research with ten-fold dilutions could provide a more precise estimate.

None of the market weight pigs that replicated virus and seroconverted developed vesicular lesions in this study. The same isolate given to 5-month-old pigs at a dose of ~1x10$^7$ TCID$_{50}$/mL did result in the development of lesions in 6/8 pigs in a previous study [22]. It remains unclear why some animals develop lesions after exposure to SVA, while others do not. Although most animals in experimental infection studies have been reported to develop lesions, this could be attributed to the high doses of virus typically used for inoculation (10$^7$–10$^8$ TCID) [9–12, 23, 24]. Reports of the incidence of vesicular lesions in the field have ranged from 10–90% of animals on affected farms [6, 25–27]. In addition, lesion development may vary based on pathogenicity of the viral isolate. A recent study compared the pathogenesis of two Chinese SVA isolates (2016, 2017), and they observed the 2016 isolate most similar to contemporary Canadian isolates did not result in any clinical disease, while the 2017 isolate similar to US strains did result in vesicular lesions in finishing pigs [28].

SVA was first detected in rectal swabs of finishing pigs from the 10$^{6.5}$ group on 1 dpi with the 10$^{4.5}$ group following on 4 dpi and finally the 10$^{2.5}$ group on 7 dpi. In addition, the duration of PCR positive rectal swabs was correlated with dose as supported by the 10$^{6.5}$ group shedding for 12 days, the 10$^{4.5}$ group around 7 days, and the 10$^{2.5}$ group for 2 days. Some of

**Table 6. Summary of SVA infection outcomes in neonatal pigs by PCR and VN assays.**

| TCID$_{50}$/mL | TCID$_{50}$/pig* | Ct | PCR positive | VN positive |
|---|---|---|---|---|
| 501,000 | 1,002,000 | 18.3 | 4/4 (100%) | 4/4 (100%) |
| 31,600 | 63,200 | 21.9 | 2/4 (50%) | 2/4 (50%) |
| 5,010 | 10,020 | 26.1 | 1/4 (25%) | 1/4 (25%) |
| 316 | 632 | 29.7 | 1/4 (25%) | 0/4 (0%) |
| 31.6 | 63.2 | 33.2 | 0/4 (0%) | 0/4 (0%) |
| 0 | 0 | 36.5 | 0/4 (0%) | 0/4 (0%) |

*Pigs were inoculated with 2mL

the same trends were also observed in the PCR results from serum and oral fluids. These PCR results support the SVA dose an animal is exposed to may affect the infection dynamics including viral replication and shedding. The inoculum dose and route of exposure of FMDV in swine affects the incubation period and infection dynamics with greater doses shortening the incubation period and decreasing time to clinical signs [29–31]. Therefore, biosecurity measures in the field to decrease viral exposure could aide in reducing the severity of infection and spread on swine farms.

The four rooms housing finishing pigs in Study 1 were reused for each challenge dose starting with the highest dilution ($10^{0.5}$) and finishing with the lowest dilution challenge group ($10^{6.5}$) due to animal housing constraints. In between challenge groups, rooms were washed down, treated with Virkon™ S and allowed to dry (repeated once), and rinsed with water. PCR negative environmental swabs of gating in the room supported the efficacy of the cleaning and disinfection process for rooms between usage. Other disinfection procedures including accelerated hydrogen peroxide and bleach have shown efficacy against SVA [32, 33]. A positive environmental swab from a room housing a pig in the $10^{6.5}$ group occurred right after the peak of SVA detection in rectal swabs; however, the remaining samples all tested negative suggesting there was not a high load of SVA in the environment of these rooms. One explanation for lack of SVA detection in the environment of this study could be individual animal housing and is not representative of the load that may be present in a barn filled with infected animals.

Field reports from SVA affected breeding herds have described increased neonatal mortality in piglets during the first week of life and the syndrome was termed epidemic transient neonatal losses (ETNL) [5, 25, 27, 34]. Clinical signs observed included lethargy, wasting, and diarrhea with cases in Brazil also reporting neurologic signs and vesicular lesions [17, 25]. Most cases of neonatal mortality in the US did not observe significant gross or histologic lesions in affected piglets [34]. In Brazil, immunohistochemistry identification of SVA in histologic lesions of the bladder, intestines, and central nervous system plus PCR testing to rule out other viral agents provided evidence that SVA was the causative agent for these lesions in neonates [15, 35]. Piglets in Study 2 were between 2–4 days-of-age when inoculated with SVA and the only clinical sign observed some piglets across all challenge groups was soft stool during the first week after infection. Although SVA cannot be ruled out in the pigs that were PCR positive for SVA, the change of diet from sow's milk to milk replacer likely played a role in the change in stool consistency. Thus, neonatal mortality due to SVA infection in breeding herds has yet to be experimentally reproduced and further research should be performed to better understand the contributing factors to ETNL.

Dose effect on acute infection dynamics in the neonates could not be evaluated in this study since small ports on the cages were only opened for feeding during individual housing to reduce the risk of virus contamination in the room and exposure of animals to virus beyond the dose received during inoculation. The negative PCR status of the sentinel pig in each challenge room supported that neonates were only exposed to the inoculum dose of SVA. The first samples collected from the neonates was on 6 dpi when animals were removed from individual housing and those were used to determine PCR positive status. PCR data collected after 6 dpi could not be used for infection status since it could be confounded by exposure to shedding from other infected animals in the group housing, but that information was critical to shedding light on the infectivity of positive pigs and how readily SVA can transmit to naïve animals.

Epidemiologic investigations conducted on SVA affected breeding farms suggested indirect transmission of SVA through contaminated farm employees, livestock trailers or carcass removal equipment were likely routes of virus introduction [25]. Environmental sampling to determine level of SVA contamination is most often tested by PCR. Limited information is

available to correlate SVA PCR Ct values to infectious virus. In this study, the minimum infectious dose for finishing pigs had a Ct value of 25.6 and neonates 29.7. Again, only hundred-fold dilutions were tested on finishing pigs, so the true minimum infectious dose may have a higher Ct value. Inoculum Ct values of 32.6 and 33.2 for finishing pigs and neonates respectively were not able to infect pigs. Environmental samples with Ct values around 32 or greater may not present a large risk for infecting pigs and spreading SVA.

This study determined the minimum infectious dose of a 2011 SVA isolate after intranasal inoculation in finishing pigs and oral inoculation in neonates. Differences in the dilution series and inoculation route could have contributed to differences in the minimum infectious dose between finishing pigs and neonates [29]. Recent studies have focused on more natural routes of exposure to determine minimum infectious dose such as oral exposure using natural feeding and drinking behaviors and pig-contact exposure [31, 36]. In addition, other SVA isolates such as more contemporary strains may have different infectious doses. Information from this work can be used in future research to make more precise estimates of the infectious dose SVA with other strains and exposure routes. Understanding the minimum infectious dose of SVA can help producers and veterinarians in the swine industry focus their disease control and bio-security measures on areas that carry the most risk of exposure of high levels of SVA.

## Supporting information

**S1 Table. Individual animal values for Fig 2.**
(XLSX)

## Acknowledgments

The authors thank Deb Adolphson and Sarah Anderson for technical assistance and Jason Huegel, Justin Miller, Keiko Sampson, Nate Horman, and Dr. Jean Kaptur for assistance with animal studies.

## Author Contributions

**Conceptualization:** Alexandra Buckley, Kelly Lager.

**Formal analysis:** Alexandra Buckley.

**Investigation:** Alexandra Buckley.

**Methodology:** Alexandra Buckley.

**Project administration:** Kelly Lager.

**Supervision:** Kelly Lager.

**Visualization:** Alexandra Buckley.

**Writing – original draft:** Alexandra Buckley.

**Writing – review & editing:** Kelly Lager.

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
