## [Decision Letter · Decision Letter 0]

4 Apr 2022

Infectious dose of Senecavirus A in market weight and neonatal pigs

PONE-D-22-06004

Dear Dr. Buckley,

We’re pleased to inform you that your manuscript has been judged scientifically suitable for publication and will be formally accepted for publication once it meets all outstanding technical requirements.

Kind regards,

Douglas Gladue, Ph.D

Academic Editor

PLOS ONE

Additional Editor Comments (optional):

One of the reviewers didn't respond, so I reviewed the paper as well, and agree with the first reviewer. Congrats on a nicely written manuscript.

Reviewers' comments:

Reviewer's Responses to Questions

**Comments to the Author**

1. Is the manuscript technically sound, and do the data support the conclusions?

Reviewer #1: Yes

2. Has the statistical analysis been performed appropriately and rigorously? 

Reviewer #1: Yes

3. Have the authors made all data underlying the findings in their manuscript fully available?

Reviewer #1: Yes

4. Is the manuscript presented in an intelligible fashion and written in standard English?

Reviewer #1: Yes

5. Review Comments to the Author

Reviewer #1: Good establishment work for the challenge of swine of different ages with SVA. Well written and reads well. While the authors did not observe vesicular lesions in the study they do establish the presence of infection and replicating virus in inoculated swine. While observed lesions would have been ideal, as the authors reference in the discussion the lack of lesions following SVA challenge is not unprecedented.

6. PLOS authors have the option to publish the peer review history of their article (what does this mean?). If published, this will include your full peer review and any attached files.

Reviewer #1: No

---

## [Editor Report · Acceptance letter]

20 Apr 2022

PONE-D-22-06004 

Infectious dose of *Senecavirus A* in market weight and neonatal pigs 

Dear Dr. Buckley:

I'm pleased to inform you that your manuscript has been deemed suitable for publication in PLOS ONE. Congratulations! Your manuscript is now with our production department. 

Kind regards, 

on behalf of

Dr. Douglas Gladue 

Academic Editor

PLOS ONE